



# NOₓ emission trends over Chinese cities estimated from OMI observations during 2005 to 2015

Fei Liu[1,2,3,*], Steffen Beirle[3], Qiang Zhang[1], Ronald J. van der A[2], Bo Zheng[4], Dan Tong[1] and Kebin He[1,4]

[1]Ministry of Education Key Laboratory for Earth System Modeling, Department of Earth System Science, Tsinghua University, Beijing, China
[2]Royal Netherlands Meteorological Institute (KNMI), P.O. Box 201, De Bilt, the Netherlands
[3]Max-Planck-Institut für Chemie, Mainz, Germany
[4]State Key Joint Laboratory of Environment Simulation and Pollution Control, School of Environment, Tsinghua University, Beijing, China
[*]now at: Universities Space Research Association (USRA), GESTAR, Columbia, MD, USA

*Correspondence to*: Fei Liu (fei.liu@nasa.gov ; liuf1010@gmail.com)
                     Qiang Zhang (qiangzhang@tsinghua.edu.cn)

**Abstract.** Satellite $NO_2$ observations have been widely used to evaluate emission changes. To determine trends in $NO_x$ emission over China, we used a method independent of chemical transport models to quantify the $NO_x$ emissions from 48 cities and 7 power plants over China, on the basis of Ozone Monitoring Instrument (OMI) $NO_2$ observations during 2005 to 2015. We found that $NO_x$ emissions over 48 Chinese cities increased by 52% from 2005 to 2011 and decreased by 21% from 2011 to 2015. The decrease since 2011 could be mainly attributed to emission control measures in power sector; while cities with different dominant emission sources (i.e. power, industrial and transportation sectors) showed variable emission decline timelines that corresponded to the schedules for emission control in different sectors. The time series of the derived $NO_x$ emissions was consistent with the bottom-up emission inventories for all power plants (r=0.8 on average), but not for some cities (r=0.4 on average). The lack of consistency observed for cities was most probably due to the high uncertainty of bottom-up urban emissions used in this study, which were derived from downscaling the regional-based emission data to cities by using spatial distribution proxies.

## 1 Introduction

Nitrogen oxides ($NO_x$), including nitrogen dioxide ($NO_2$) and nitric oxide (NO), are atmospheric trace gases with a short lifetime, and they actively participate in the formation of tropospheric ozone and secondary aerosols and thus harm human health and significantly affect climate (Seinfeld and Pandis, 2006). Anthropogenic activities, particularly fossil fuel consumption, are the most important sources of $NO_x$ emissions. Anthropogenic $NO_x$ emissions are clustered over densely populated urban areas and suburban/rural industrial areas where large point sources such as power plants are located.

Tropospheric $NO_2$ observations detected from space have been applied to infer the strength of $NO_x$ emissions. The concentration of $NO_2$ in a vertical column of air can be measured via satellite instruments and related to $NO_x$ emissions according to the mass balance by considering transport and chemical conversion. A pioneering study has used the downwind decay of $NO_2$ in continental outflow regions to estimate the average $NO_x$ lifetime and global $NO_x$ emissions (Leue et al., 2001). Subsequent studies have used chemical transport models (CTMs) to exploit satellite measurements as a constraint to improve $NO_x$ emission inventories at the global/regional scale (e.g., Martin et al., 2003; Konovalov et al., 2006; Kim et al., 2009; Lamsal et al., 2011). The spatial and temporal resolution of tropospheric $NO_2$ observed from space has increased over time, from the Global Ozone Monitoring Experiment (GOME), which was launched in 1995 (Burrows et al., 1999), to the Ozone Monitoring Instrument (OMI) (Levelt et al., 2006), which was launched in 2004 and enables the use of satellite retrievals to resolve emissions at a finer scale. OMI $NO_2$ observations sorted according to wind direction from wind fields developed by





the European Center for Medium range Weather Forecasting (ECMWF) have been fitted by Beirle et al. (2011), who have used the exponentially modified Gaussian function, which allows for a simultaneous fit of the $NO_x$ lifetime and emissions for megacities without further input from CTMs. In the previous work, we have advanced this method for estimating $NO_x$ emissions from sources located in a polluted background (Liu et al., 2016a). An alternative approach to quantifying urban $NO_x$ emissions, proposed by Valin et al. (2013), involves rotating satellite observations according to wind directions such that all observations are aligned in one direction (from upwind to downwind), thus increasing the number of observations. Subsequent studies have applied the concept of CTM-independent methods for estimating $SO_2$ by introducing an advanced three-dimensional function (Fioletov et al., 2015; Fioletov et al., 2016; McLinden et al., 2016).

Satellite observations are particularly suitable for evaluating emission changes because they provide continuous and timely tropospheric $NO_2$ measurements with global coverage (Lelieveld et al., 2015). Changes in the spatial heterogeneity of $NO_2$ trends have been observed worldwide, and substantial decreases over Europe and the US (Russell et al., 2012) and significant increases over Asia have widely been detected in recent decades (Richter et al., 2005). A linear function superposed on an annual seasonal cycle has been introduced by van der A et al. (2008) to derive a quantitative estimate of emission trends for a grid with a spatial resolution of $1° \times 1°$ by fitting the corresponding monthly $NO_2$ columns. Follow-up studies (e.g., Schneider and van der A, 2012; Schneider et al., 2015) have applied similar statistical analyses to time series of $NO_2$ in finer grid cells located over the center of the city and have quantified the long-term average pattern of $NO_2$ for megacities. The multi-annual (moving) average is an alternative method of describing local $NO_2$ trends. The interannual variation in the mass of a chemical species integrated around the source has been used as an indicator of emission changes and has been shown to be capable of illustrating the emission changes over US power plants (Fioletov et al., 2011), Canadian oil sands (McLinden et al., 2012) and Indian power plants (Lu et al., 2013). In addition, de Foy et al. (2015) and Lu et al. (2015) have adopted the fitting function proposed by Beirle et al. (2011) and have provided estimates of $NO_x$ emission trends from isolated power plants and cities over the US on the basis of 3-year average $NO_2$ values obtained through the plume rotation technique described by Valin et al. (2013).

China is one of the largest $NO_x$ emitters in the world and is the source of approximately 18% of the global $NO_x$ emissions (EDGAR 4.2, EC-JRC/PBL, 2011). China has experienced rapid increases in $NO_x$ emissions because of its growing economy over the past two decades, during which emissions have increased by a factor of three (Kurokawa et al., 2013) and have caused severe air pollution. To improve air quality, the Chinese government implemented new emission regulations aimed at decreasing the national total $NO_x$ emissions by 10% between 2011 and 2015 (The State Council of the People's Republic of China, 2011). Several recent studies (e.g., Duncan et al., 2016; Krotkov et al., 2016) have suggested the effectiveness of the air quality policy, as evidenced by a decreasing trend in $NO_2$ columns over China since 2012. Miyazaki et al. (2017) and van der A et al. (2017) have further reported a recent decline in national $NO_x$ emissions on the basis of satellite data assimilation. Liu et al. (2016b) have studied changes in $NO_2$ column densities for each province from 2005 to 2015 and have performed an intercomparison of a bottom-up inventory and satellite observations; the study attributes the decline in regional $NO_2$ to decreased emissions from power plants and urban vehicles. However, few analyses have been performed for individual cities or power plants, which are the primary targets of the new control measures. Such investigations may provide stronger evidence of the effects of control measures on $NO_x$ emissions.

In this work, we quantified $NO_x$ emission trends over urban areas in China from 2005 to 2015. Certain widely used approaches, including linear trend analysis (e.g., Duncan et al., 2016; Krotkov et al., 2016) and exponentially modified Gaussian method (e.g., de Foy et al., 2015; Lu et al., 2015), are difficult to directly apply to hot spots in China. The linear trend analysis approach is particularly useful for quantifying changes for cities with a linear trend; however, it is not applicable to most Chinese cities, which show a clear turning point of emissions. The exponentially modified Gaussian method may introduce significant uncertainties to the fit results because of the heterogeneously polluted background over China (de Foy et al., 2014; Liu et al., 2016a). We applied our advanced fitting function to sources located in a polluted background (Liu et al., 2016a) to calculate





the 3-year moving averages of NO$_x$ emissions of pollution hotspots including individual cities and power plants, and to relate their variations to bottom-up information. The main purpose of this study was not only to demonstrate the recent decrease in NO$_2$ levels across the country, as indicated by previous reports (Liu et al., 2016b) but also to display the diverse emission characteristics among cities and provide in depth interpretations of these characteristics. The fitting function and data sets used

in this study are detailed in Sect. 2. The interannual variations of NO$_x$ emissions and the analysis of emission trends for cities derived from the fitting function are provided in Sect. 3.1 and Sect. 3.2, respectively. The fitting results for cities are presented in Sect. 3.3. The uncertainties associated with the fitting results are discussed in Sect. 3.4, and the primary findings of this study are summarized in Sect. 4.

## 2 Methodology

### 2.1 Fitting method

We improved the exponentially modified Gaussian method (Beirle et al., 2011) to quantify the multi-year average NO$_x$ emissions obtained from OMI NO$_2$ observations for sources located in a polluted background (Liu et al., 2016a). In this work, we adapted the fitting functions of Liu et al. (2016a) to calculate the NO$_x$ emissions for individual cities and power plants, including adjustments to meet the requirements of the trend analysis.

Consistently with our previous study (Liu et al., 2016a), we used the OMI tropospheric NO$_2$ (DOMINO) v2.0 product (Boersma et al., 2011) together with the ECMWF ERA interim reanalysis (Dee et al., 2011) to perform the analysis. We calculated the mean NO$_2$ tropospheric vertical column densities (TVCDs) for calm wind speeds below 2 m/s and 8 different wind direction sectors, by following the approach in Beirle et al. (2011), and for weak-wind conditions (below 3 m/s), by following the recommendations in Lu et al. (2015), from 2005 to 2015. We used only non-winter data (from March to

November) because these data should have larger uncertainties because of larger solar zenith angles and variable surface albedo (snow). In addition, the longer NO$_x$ lifetimes in winter resulted in a less direct relationship between NO$_x$ emissions and satellite NO$_2$ observations.

Emissions were derived in a two-step approach in Liu et al. (2016a). The first step was to use NO$_2$ patterns under calm wind conditions as a proxy for the spatial distribution of NO$_x$ emissions and determine the effective atmospheric NO$_x$ lifetime from

the change of spatial patterns measured at higher wind speeds. The second step was to derive emissions from the NO$_2$ mass integrated around the source of interest divided by the corresponding lifetime.

To perform a trend analysis, we adjusted the method as follows: we based the estimation on NO$_2$ columns around the source of interest averaged over three years, in agreement with previous studies (e.g., Fioletov et al., 2011; Lu et al., 2015); and then the total NO$_x$ mass was integrated over the mean TVCDs at weak wind speeds (below 3 m/s) instead of calm winds to balance

the need for increasing the number of observations and minimizing interferences by advection. Notably, we were not able to derive valid lifetimes on the basis of the 3-year average NO$_2$ columns; instead, we fitted the lifetimes on the basis of multiple-year data (the entire study period) because of the lack of sufficient observations for different wind sectors within a 3-year period. Therefore, the NO$_x$ emissions for each 3-year period were calculated by dividing the corresponding total NO$_x$ mass by the multiple-year average lifetime. In this way, the temporal variations in NO$_x$ emissions were merely dependent on the changes

in the total NO$_x$ mass, excluding background pollution, assuming that the lifetimes did not change over time. However, we wanted to include the fit over the lifetime in this study to make the comparison of top-down and bottom-up estimates more straightforward. Subsequently, we included mountainous sites, which were defined as sites where the absolute difference in elevation between ECMWF and GTOPO data (available at https://lta.cr.usgs.gov/GTOPO30, rescaled to 0.05°) was larger than 250 m, in the following analysis. Our previous findings (Sect. 2.6, Liu et al., 2016a) have indicated that appropriate wind fields,

which are required for accurate lifetime calculations, may not always be achieved from the ECMWF simulation over mountainous regions. However, depending on changes in the total NO$_x$ mass, the fitted emission trends are not as sensitive as





the fitted lifetimes to wind fields; thus, we did not exclude mountainous sites from the trend analysis. The fitting results with poor performance (e.g., R<0.9, large confidence interval CI) were discarded, in accordance with the criteria in Sect. 2.2 of Liu et al. (2016a).

We selected the Huolin power plant (site 2#, 45.5°N, 119.7°E), which is located in Holingol, a county-level city of Inner
Mongolia of China (shown in Fig. 1), to demonstrate our approach. The Huolin power plant has a total capacity of 2400 MW and dominates the $NO_x$ emissions from the city of Holingol, contributing over 80% of the total emissions estimated by using the Multi-resolution Emission Inventory for China (see Sect. 2.2), which is a bottom-up emission inventory. Fig. 2(a) displays the 3-year average $NO_2$ TVCDs around the power plant under weak-wind conditions from 2005 to 2015. For simplicity, the 3-year period is represented by the middle year with an asterisk (e.g., 2006* denotes the period from 2005 to 2007). A
significant increase in TVCDs was observed from 2006* to 2010*, which was followed by a subsequent decrease. Fig. 2(b) presents the fitted background and $NO_x$ emissions. The fitted $NO_x$ emissions showed an increase of up to a factor of four from 2006* to 2010* and a decrease of 30% from 2011* to 2014*, whereas the fitted background was steady and showed a standard deviation of less than 10% from 2006* to 2014*. The growth of the fitted $NO_x$ emissions in the early stage was found to be consistent with the construction of new electric-generating units, with the total capacity increasing from 300 MW to 2400 MW
from 2005 to 2009. Subsequently, the fitted $NO_x$ emissions remained steady from 2010* to 2012*, when no new electric-generating units were placed into service, and finally decreased after the installation of Selective Catalytic Reduction (SCR) equipment at the power plants. This decrease in emissions indicated the effectiveness of SCR equipment for decreasing emissions.

### 2.2 Bottom-up information

We used bottom-up information to pre-select promising sites and to perform a comparison with the fitted top-down emission trends. We used the MEIC (http://www.meicmodel.org), compiled by Tsinghua University, for urban emissions. Vehicle population and coal consumption at the city level were derived from the China Statistical Yearbook for Regional Economy (NBS: CSYRE, 2004–2014) and the China Environment Yearbook (NBS: CEY, 2004–2015), respectively. We derived the information for the coordinates, unit capacities, technologies and emissions for individual power plants from the unit-based
China coal-fired Power plant Emissions Database (CPED) (Liu et al., 2015).

We calculated the $NO_x$ emissions from cities and power plants by using the MEIC and the CPED inventories, respectively, from 2005 to 2015. Only emissions for non-winter seasons were considered, in accordance with the emissions included for the top-down estimates. The gridded MEIC inventory was integrated over a $40 \times 40$ km$^2$ metropolitan area for which the proposed top-down method was sensitive to calculate the total urban emissions (Liu et al., 2016a). Notably, the emissions uncertainties
associated with power plants were much lower (30%) than those for cities (50%~200%) because the former was calculated directly from unit-level information whereas the latter was derived by downscaling regional-based emission data to finer grids using spatial proxies and integrating emissions from corresponding grids.

### 2.3 Selection of locations

We selected large cities and power plants over China as the pre-selected sites for which bottom-up emission information was
derived from the and MEIC and CPED inventories, respectively. China classifies its administrative divisions into five practical levels (from large to small): province, prefecture, county, township and village. Only prefecture-level cities were selected for analysis in this study. Power plants located in a radius of 100 km around prefecture-level city centers, on the basis of a visual inspection of satellite imagery from Google Earth, were excluded. Power plants with $NO_x$ emission rates greater than 10 Gg/yr were selected for emission fitting. Fig. 1 illustrates all investigated sites where the fit results showed good performance (see
Sect. 2.2 for details). Among over 200 pre-selected cities, 48 cities (including 14 mountainous sites) were fitted with





satisfactory results. While among over 100 pre-selected power plants, only 7 power plants (including 3 mountainous sites) were fitted with satisfactory results. Detailed information on the sites is tabulated in Table S1 of the supplement.

## 3 Results and Discussion

### 3.1 Interannual trends in OMI $NO_x$ emissions for cities

The trends in the fitted $NO_x$ emissions for 48 cities from 2006[*] to 2014[*] are shown in Fig. 3a, with an average growth trend of 52% prior to 2011[*] for all investigated cities and a declining trend of 21% from 2011[*] to 2014[*]. The $NO_x$ emissions over urban areas essentially represent a marker for combustion-related emissions, including coal combustion for power generation and industrial processes and oil combustion for transportation. Fig. 3b further summarizes the statistical data of industrial coal consumption (open squares) and vehicle population (proxy of oil consumption, solid squares), which are available for only 28

cities. Not surprisingly, the fitted emission trends for the 28 cities (circles) were consistent with those for the overall 48 cities. The observed sharp growth of 47% in $NO_x$ emissions during the period of 2006[*]–2011[*] was attributed to the growth of 75% and 158% in coal consumption and vehicle population, respectively. Coal consumption and vehicle population continued to rise and increased by 8% and 26% from 2011[*] to 2014[*] respectively; however, a subsequent decline in $NO_x$ emissions was observed. We further divided these $NO_x$ emissions by coal consumption and vehicle population because their respective

temporal variations can be treated as an approximation of the average emission factor trends in the industrial (including power) and transportation sectors, on the basis of the assumption that contributions of $NO_x$ emissions from corresponding sectors are constant over time. The ratio of emissions to coal consumption and vehicle population has diminished over time and decreased by 37% and 65% from 2011[*] to 2014[*], respectively. This declining trend was greater than the fluctuations in contributions of $NO_x$ emissions from the corresponding sectors (ranging from 1%–24% for individual cities) and indicated the effectiveness of

emission control measures.

The changes in $NO_x$ emissions from 2005 to 2015 according to sector for the investigated cities on the basis of MEIC estimates are summarized in Fig. 3a and indicate the driving force underlying the emission changes. In agreement with previous findings (Liu et al., 2016b), power plants are the primary component responsible for the decline in $NO_x$, and the associated bottom-up $NO_x$ emissions decreased by 59% between 2011 and 2015. This finding was further supported by the power plant emission

trends shown afterwards in Sect. 3.3. The decrease in both fitted and bottom-up emissions has accelerated since 2013 because of the implementation of air pollution prevention and control action plans (the State Council of China, 2013). Such plans require the deployment of denitration devices for coal-fired boilers and cement precalciners, and the requirements are not limited to power plants, as observed in earlier policies. By 2015, 92% of the power plant boilers and new dry-process cement kilns in China had installed denitration devices. In addition, low-efficiency small coal-fired boilers and even complete factories

have been phased out. Iron, steel and cement factories with an overall production capacity of 86 Gg, 44 Gg, and 263 Gg were shut down in China from 2013 to 2015. Additionally, Chinese cities have pursued a reduction in coal consumption through the gradual transformation of the energy system from coal to renewable energy and natural gas. For instance, Beijing has outlined plans for "coal-free zones" that ban coal usage, and these plans required the replacement of all coal-fired boilers with natural gas in inner suburban districts by 2015 (Clean Air Action Plan 2013–2017, Beijing Municipal Government, 2013). Accordingly,

China reached peak coal consumption in 2013, and a decline of 4% in coal consumption for the investigated cities was observed between 2013 and 2014 (Fig. 3b). Moreover, Chinese cities have been required to meet more stringent vehicle emission standards. For instance, Euro IV emission standards were widely implemented in 2015, and the $NO_x$ emission factor is only 2.6% of the Euro 0 standard for gasoline vehicles (Huo et al., 2012). Because of the notable success of emission control induced by stricter emission standards, the contributions of high-emitting old vehicles (Euro 0 in most cases) to overall emissions are

becoming increasingly significant. Reports have indicated that Euro 0 vehicles accounted for more than 50% of the total vehicle emissions in China in 2009 (MEP, 2010). Thus, China has marked high-emitting vehicles with yellow labels, implemented





traffic restrictions and subsidized scrappage programs for these vehicles (Wu et al., 2017). A total of 15 million yellow-label vehicles were scrapped between 2013 and 2015. Significant progress in controlling vehicle emissions has also been observed with improvements in vehicular fuel combustion efficiency and license registration control policies, which allocate quotas for new vehicles through public auction or lottery.

## 3.2 Interannual trends of NO$_x$ emissions for individual cities

The fitted results allowed for a closer examination of the trends and causes of emission changes at the individual city level instead of at a regional level, as performed in previous studies (e.g., Liu et al., 2016b). Fig. 4 compares the fitted and bottom-up emissions for selected cities, which can be considered in 3 broad categories: mega cities with large amount of vehicle emissions (Guanzhou and Shanghai in Fig. 4a and b); cities with power plants as the dominant emission source (Wuhai and Huainan in Fig. 4c and d); and cities with industrial plants as the dominant emission source (Karamay and Jiayuguan in Fig. 4e and f).

Fig. 4a and b show that megacities reached the emission peak prior to the average timeline shown in Fig. 3. Here, we discuss in detail the temporal variations in Guangzhou, the largest city in South China. The early decline in emissions was primarily related to the stricter regulations on vehicles, which was the only source that showed decreasing emissions, as indicated by the bottom-up inventory. Guangzhou implemented Euro III emission standards for all light-duty vehicles and heavy-duty diesel vehicles in 2006, which was two years earlier than the national requirement. Traffic restrictions for motorcycles and trucks and for yellow-label vehicles have also been implemented since 2007 and 2008, respectively. In addition, alternative fuels in buses and taxis have also been promoted in Guangzhou, and 75% and 94% of these vehicles, respectively, were using liquefied petroleum gas by 2009 (Zhang et al., 2013). The early decline before 2010 in Shanghai shown in Fig. 4b is attributable to similarly strict regulations for vehicle emissions that were implemented before the national schedule. In addition, Guangzhou has gradually phased out the high-pollution iron and steel industry since 2008; however, such emission reductions were not well presented by the bottom-up inventory. In line with the national denitration procedure, coal-fired power plants have remained a significant contributor to emission reductions since 2011. The bottom-up NO$_x$ emissions from power plants in Guangzhou decreased by over 50% between 2011 and 2015, because of the wider deployment of denitration devices at power plants.

The interannual trends of NO$_x$ emissions for the cities of Wuhai and Huainan are displayed in Fig. 4c and d, and power plants were the dominant source of NO$_x$ emissions. Not surprisingly, the top-down and bottom-up information was more consistent than the information for the other two categories, because of the better quality of emission estimates for the power sector. The fitted emissions decreased with the decline in emissions from power plants around 2012, and this finding was related to the deployment of denitration devices.

Emission variations for cities for which the industrial sector was the dominant emission source are shown in Fig. 4e and f, which indicate significant inconsistencies in the top-down and bottom-up information, even for the total amount. Cities belonging to this category were usually medium and small cities. For instance, the city of Jiayuguan (Fig. 4f) has a total population of 0.2 million, a vehicle population of 0.03 million and a large-sized industrial enterprise (Jiuquan Iron & Steel (Group) Co., Ltd). Industrial activities are the most likely contributor to the recent deceleration and even decline in total emissions, because of the small human and vehicle populations and the limited amount of power plant emissions (light blue line) in the city. To meet the demands of the air pollution prevention and control action plan (the State Council of China, 2013), the iron and steel enterprises have been required to regulate their emissions since 2013. Additionally, the city is required to meet stricter vehicle emission standards and retire aged vehicles. However, the bottom-up inventory for the city of Jiayuguan was consistent with this analysis for only the transportation sector and not the industrial sector as shown in Fig. 4f. The bottom-up transportation emissions experienced a decline of 10% and a sharp increase of 20% in the vehicle population between 2013 and 2015. In addition, the NO$_x$ emissions from the industrial sector were fairly steady and showed a decrease of only 2%



between 2013 and 2015 and account for a small share (20%) of the total emissions. Such changes in the industrial sector most probably represent the regional average level and do not represent the levels for a city with a large-sized iron and steel enterprise, because of the uncertainty of downscaling approaches adopted in the bottom-up inventory.

Although we used bottom-up inventories to interpret the changes in $NO_x$ emission, certain notable discrepancies occurred

between the fitted emissions and the bottom-up inventories. We further explored the reasons for these inconsistencies in cities by examining the differences in trends between top-down and bottom-up estimates at different spatial scales. Fig. 5 presents the temporal variations at provincial and city scales from 2006* to 2014* for the top-down and bottom-up data sets. The top-down information at the provincial level (Fig. 5a) was the 3-year average OMI $NO_2$ column densities for non-background regions, where the average annual $NO_2$ column densities were larger than $1 \times 10^{15}$ molec/cm$^2$ or the average $NO_2$ column

densities for summer exceeded those for winter in Liu et al. (2016b), and the top-down estimates (Fig. 5c) at the city level were derived from this study. The bottom-up emissions were calculated by summing the emissions of the corresponding grids belonging to individual provinces (Fig. 5b) or cities (Fig. 5d, see Sect. 2.2). Not surprisingly, the comparison of the two independent data sets showed that the trends at both the provincial and city levels were generally consistent, with both levels experiencing a sharp rise before 2012* (2011* in Fig. 5c) and a continuous decline thereafter. However, a closer examination

of the magnitude of relative changes showed that the differences were scale dependent. The provincial-level comparison showed a growth trend of 40% ± 26% and 34% ± 21% from 2006* to 2012* and a subsequent declining trend of 9% ± 4% and 14% ± 6% from 2012* to 2014* for the top-down and bottom-up data sets, respectively, and indicated the acceptable accuracy of provincial totals in bottom-up estimates. However, the city-level comparison exhibited a large discrepancy in the magnitude of change rates. For instance, the top-down growth rates reached 45% ± 46% in the period from 2006* to 2012*, whereas the

bottom-up rates were only 25% ± 27% for the same period.

We expect that the scale dependence of the differences shown in Fig. 5 may be explained by the spatial allocation approach adopted in bottom-up inventories. Current gridded bottom-up emission inventories rely heavily on spatial proxies because rare emissions, excluding the emissions from stacks of large point sources, can be directly measured. A variety of spatial proxies, such as population density, road density and satellite-observed nightlights, are used to geographically distribute emission totals

from a large scale down to the scale of geographic grids of various sizes. Several studies (e.g., Hogue et al., 2016) have indicated that such a spatial distribution approach using proxy data introduces significant uncertainties because emissions can be misallocated spatially and temporally. Although the MEIC inventory has substantially improved its accuracy, such as by using the high-resolution power plant database CPED (Liu et al., 2015), a lack of data has led to the inclusion of other types of point sources (such as industrial boilers) as areal sources of emissions. For example, the MEIC first downscales provincial

industrial emission totals to county totals according to industrial GDP values and then distributes county emissions to grids according to the population density. However, industrial facility locations are likely to be decoupled from spatial proxies, because polluted facilities are often required to be located in rural areas with smaller GDP and populations (Zheng et al., 2017), and this decoupling may have resulted in the underestimation of emissions from steel and iron factories shown in Fig. 4f. In addition, the spatial distribution of proxies cannot easily represent the emission changes caused by anti-leapfrogging policies

implemented in cities ahead of the national schedule, such as the previously discussed new vehicle emission standards in Guangzhou. Thus, regional diversity may have been diminished and consequently resulted in the small standard deviation over cities shown in Fig. 5d.

The correlation coefficients of the pair-wise trends between the fitted $NO_x$ emissions and the bottom-up inventory for the period 2006* to 2014* are illustrated in Fig. 6. The correlation coefficient of the time series of $NO_x$ emissions showed

remarkable diversity for cities and reached over 0.9 for Urumqi (#9 in Fig. 6) and dropped to less than -0.7 for Jinzhou (#5 in Fig. 6), probably because of the high uncertainties of the bottom-up inventories for cities. Notably, the negative correlation coefficients do not necessarily correspond to a strong inverse linear relationship and may suggest inconsistency over only one or two periods (Fig. 4b). Additionally, the negative correlation coefficients were always observed when the time series of fitted



emissions experienced a minor fluctuation without a significant trend, as demonstrated in Fig. 6b by cities with a correlation coefficient less than -0.4.

### 3.3 Interannual trends in OMI NO$_x$ emissions for power plants

The trends in fitted NO$_x$ emissions for 7 power plants from 2006[*] to 2014[*] are shown in Fig. 7. The changes in the total NO$_x$

mass and derived NO$_x$ emissions were consistent with the addition of new units in individual power plants until the installation of denitration devices. The dramatic growth in NO$_x$ emissions (red line) prior to 2010[*], which reached 89% on average for all power plants investigated in this study, was driven by increases in the capacity of 84% for the corresponding power plants (gray bar). However, the subsequent decline in NO$_x$ emissions could not be explained by the simultaneous changes in total unit capacities, which increased by 3% from 2010[*] to 2014[*], but suggests a good agreement with the wider deployment of

denitration devices, such as SCR equipment. The installation of SCR devices generally ensures a NO$_x$ removal efficiency of 80–85% (Forzatti et al., 2001). However, the denitration devices used in Chinese power plants usually do not meet this standard efficiency, because of the non-optimal use of catalysts and reductants. The average removal efficiency of SCR equipment for 2014 was only 60% on the basis of statistics from the CPED. In this way, the increasing market share of SCR equipment (up to 73%, blue line) corresponded to a decrease of approximately 40% (i.e., 73% $\times$ 60%) in NO$_x$ emissions, a result consistent

with the changes in fitted emissions. The fitted emissions were further compared with the bottom-up emission estimates, and both values shared a similar trend. The significant decline of $40 \pm 22\%$ (mean $\pm$ standard deviation) in fitted NO$_x$ emissions for individual power plants from 2010[*] to 2014[*] was generally consistent with the simultaneous decline in bottom-up estimates of $22 \pm 29\%$. However, a minor difference in the peak year of emissions was detected for a few power plants, and was most probably caused by uncertainty in the fitted emissions related to the lack of interannual variations in NO$_x$ lifetimes.

China has implemented the new emission standards for thermal power plants (Ministry of Environmental Protection of China (MEP), 2011) in 2012, requiring power plants, particularly large plants, to install denitration devices, such as SCR equipment, to control their NO$_x$ emissions. The deployment of denitration devices (shown in Fig. 7) was consistent with this new policy, and the national average market share of SCR equipment grew from 18% to 86% between 2011 and 2015 (China Electricity Council, 2012–2016). Given that the overall capacity of the power plants investigated in this study was equivalent to only 2%

of the total national capacity; we may not able to conclude that the temporal variations in NO$_x$ emissions derived from the 7 power plants reflect the emissions from large power plants at the national level. While for the investigated power plants, the derived emissions were consistent with the bottom-up emissions and the time series of the two estimates were well correlated, even for mountainous sites where the absolute values of the emission estimates differed significantly (Fig. 6a). The good consistency increased our confidence that the fitted emission trends accurately represented the real-word emission variations,

because the uncertainty of the bottom-up emission inventory for power plants is fairly low (Liu et al., 2015).

### 3.4 Uncertainties

The fitted NO$_x$ emissions were compared with the bottom-up emission estimates (Sect. 2.2) for all 48 cities and 7 power plants in Fig. 8, and their correlations were consistent with the average emission estimates for multiple years shown in the previous work (Liu et al, 2016a). In general, the comparisons indicated consistency among non-mountainous sites, which presented a

higher correlation coefficient for power plants (blue symbols in Fig. 8a, r=0.89) than cities (blue symbols in Fig. 8b, r=0.81). The results for the mountainous sites showed higher scatter for both power plants (red symbols in Fig. 8a, r=0.79) and cities (red symbols in Fig. 8b, r=0.44), thus confirming that those top-down estimates had higher uncertainties because of inaccurate ECMWF wind fields for mountainous sites (Liu et al, 2016a). The comparable correlation among the results presented here and in the previous study by Liu et al (2016a) increased our confidence in the accuracy of the fitted results.

We estimated the uncertainty of the fitted NO$_x$ emissions and their trends by using a method analogous to that in Liu et al. (2016a) because of the consistency in methodology between the two studies. The uncertainty analysis was performed on the




basis of the fit performance and according to sensitivity studies that have investigated the dependencies on a priori settings, which are detailed in the supplement of Liu et al. (2016a). The major sources of errors contributing to the overall uncertainties included (a) fit error; (b) choice of fit intervals; (c) tropospheric $NO_2$ VCDs and the $NO_x/NO_2$ ratio; (d) choice of wind fields and (e) lifetime variations. The uncertainties arising from (a) – (c) were consistent with those reported in Liu et al. (2016a).

Here, we briefly discuss the impact of (d) and (e).

(d) Choice of wind fields. The $NO_2$ trends observed under weak-wind conditions may vary from those under all-wind conditions (Lu et al., 2015), because higher wind speeds are expected to cause longer $NO_x$ lifetimes because of the faster dilution of $NO_x$ (Valin et al., 2013). A change in the weak-wind conditions by all-wind conditions affects the resulting total mass by approximately 10% on average.

(e) Lifetime variations. We use multiple-year average lifetimes and 3-year average $NO_2$ masses to calculate $NO_x$ emissions trends in this study. The variations in total $NO_2$ mass do not necessarily correlate linearly with $NO_x$ emissions, because of changes in the $NO_x$ lifetimes related to variations in meteorology and $NO_x$ chemistry. However, the temporal variations in lifetimes corresponding to the 3-year moving averages of TVCDs are reduced significantly, as supported by the similar decreases in the 3-year mean $NO_x$ emissions and OMI $NO_2$ observations over urban areas in the US (Lu et al., 2015). In

addition, we could not unambiguously relate the variability of fitted $NO_x$ lifetimes to $NO_2$ levels (Liu et al., 2016a).

The method was applied to the period prior to the row anomaly (the 3-year period from 2005 to 2007), which had a larger number of observations than the other periods. The method was successful for 19 sites, and the fitted lifetimes were not sensitive to the $NO_2$ changes within the studied period, with the lifetimes increasing by only 9% when the average $NO_2$ increased by ~20% compared with the multiple-year level. The uncertainties caused by lifetime variations were estimated to

be 10%, and this value was applied to all considered sources.

The total uncertainty was defined as the root of the quadratic sum of the aforementioned contributions, which were assumed to be independent. We estimated that the total uncertainties of the fitted $NO_x$ emissions were within 66%–99% for all investigated sites. Notably, this estimate is rather conservative because of the assumption that all the contributors to uncertainties are independent. In addition, the uncertainty in emission trends was significantly lower than that of emissions

because the errors from the choice of fit intervals, wind fields, tropospheric columns and $NO_x/NO_2$ ratios were generally compensated for in the assessment of trends.

## 4 Conclusions

We quantified the $NO_x$ emissions of cities over China obtained from satellite $NO_2$ observations for the period 2005 to 2015. The lifetimes were determined from the average changes in $NO_2$ distributions under windy conditions compared with calm

conditions, and the emissions were subsequently estimated by dividing the total mass of $NO_2$ integrated around the source of interest in any three consecutive years from 2005 to 2015 by the derived lifetimes. The method was successfully applied to 48 cities and 7 power plants to obtain the $NO_x$ emission trends over China.

We detected similar temporal variations in the derived $NO_x$ emissions for cities and power plants, both of which experienced a rapid growth until approximately 2011 and a sharp decline thereafter. The $NO_x$ emissions from selected cities experienced

an average growth of 52% prior to 2011[*], because of the increase in fuel consumption. The subsequent decline of 21% was quantitatively attributed to the successful control of $NO_x$ emissions in the power, industrial and transportation sectors. In addition to installing denitration devices at power plants and cement plants, China has transformed its industrial structure by phasing out heavily polluting industrial factories, decreasing coal consumption, controlling vehicle emissions through stricter emission standards and scrapping aged vehicles.

We further compared the derived $NO_x$ emissions with the bottom-up emission estimates for individual cities. Megacities with a large amount of vehicle emissions reached the emission peak prior to the average timeline, because of the stricter vehicle





regulations that were implemented ahead of the national schedule. Cities with power and industrial sectors as the dominant emission sources reached the emission peak at dates that were consistent with the schedule for emission control in the corresponding sectors. In addition, we found that the derived $NO_x$ emissions were significantly less consistent with the regional inventory MEIC for cities (r=0.4 on average) than the high-resolution power plant inventory CPED, a result related to the

uncertainties in the spatial allocation technique, in which surrogates were used to break down regional-based emission data to the level of cities. However, the discrepancy was strongly scale dependent, and the trends between the top-down and bottom-up estimates were consistent at the province level but not at the city level. This finding indicated that the allocation technique used in bottom-up inventory misrepresents the spatial and temporal patterns for emissions over cities.

Our results indicated that OMI $NO_2$ observations can be used to estimate $NO_x$ emission trends for individual cities and power

plants, even those with a polluted background. Moreover, this method can be applied to quantify the emission variations from various hot spots worldwide. Notably, the lifetimes were derived on the basis of the average $NO_2$ columns for the entire study period of 2005–2015 because of a lack of statistics for shorter periods. Because future satellite instruments, such as TROPOMI (Veefkind et al., 2012), GEMS (Kim et al., 2012), TEMPO (Chance et al., 2012) and Sentinel-4 (Ingmann et al., 2012), have improved spatial and temporal resolution, the capabilities of this method is expected to be further enhanced. We expect that

future estimates of interannual lifetimes as well as diurnal cycles from geostationary satellites will be able to account for changes in meteorology and $NO_x$ chemistry. In addition, the trend analysis for annual and even seasonal $NO_x$ emissions should be achievable and should serve as a more reliable tool for interpreting emission changes.

### Acknowledgements

This research was funded by the National Natural Science Foundation of China (41625020), China's National Basic Research

Program (2014CB441301), and the MarcoPolo project of the European Union Seventh Framework Programme (FP7/2007-2013) under Grant Agreement number 606953.

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





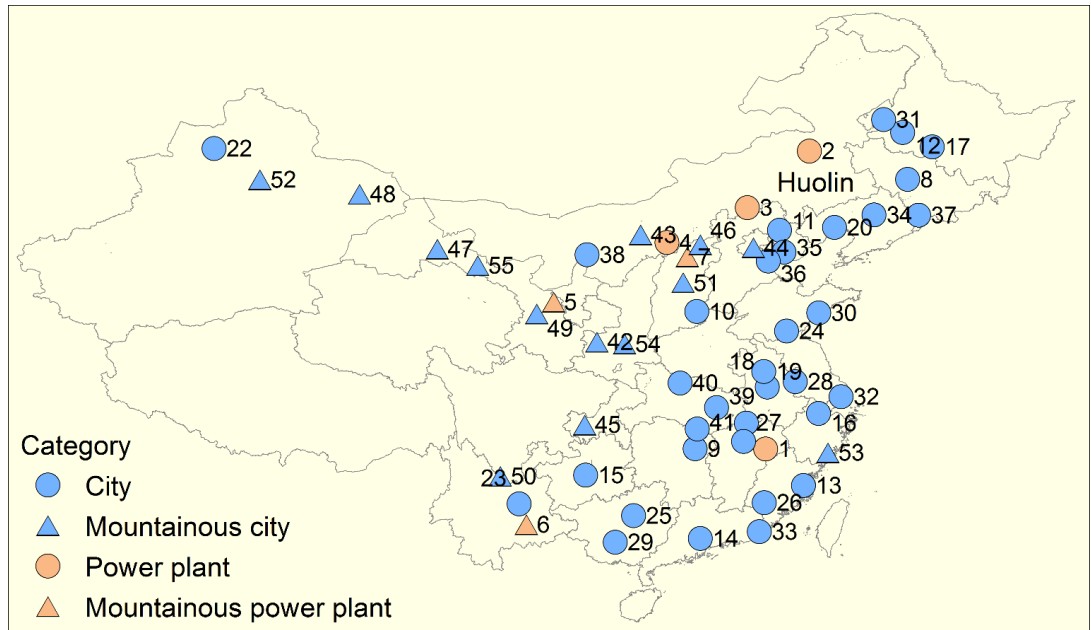

**Figure 1: Locations of the selected sites in this study. The triangles represent the mountainous sites defined in Sect. 2.1. All locations are labeled with their IDs (see Table S1).**





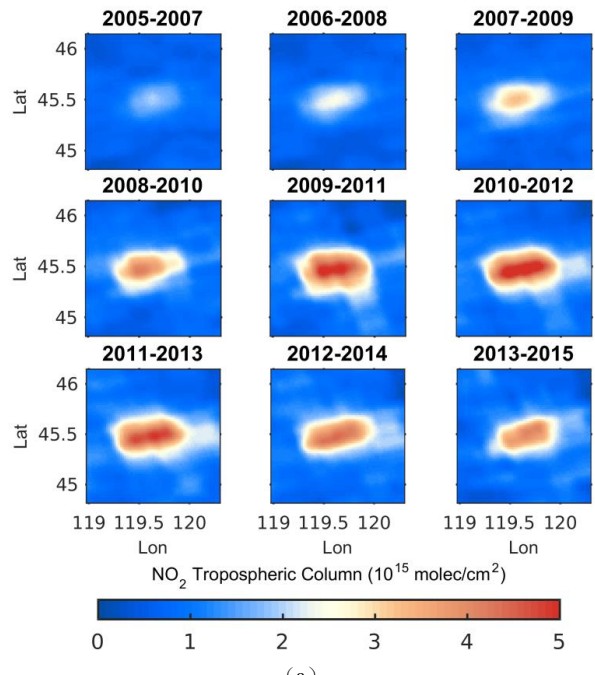

(a)

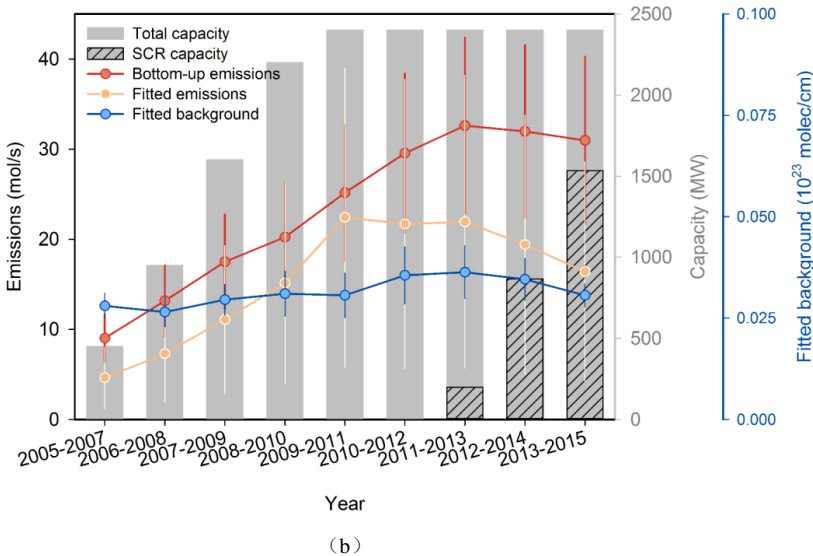

(b)

**Figure 2: (a) OMI NO₂ TVCD map under weak-wind conditions (<3 m/s) around the Huolin power plant (#2 in Fig. 1) during 2005 to 2015 and (b) the corresponding fit results. The red and blue lines denote the fitted emissions and background, respectively; the pink line denotes the bottom-up emission estimates; the solid and dashed bars denote the total capacity of the generation units and the capacity of generation units that installed SCR equipment, respectively. The information on the capacity and SCR equipment is derived from the CPED database (Liu et al., 2015). Error bars show the uncertainties for emissions by using this method and bottom-up inventories (see Sect. 3.4).**





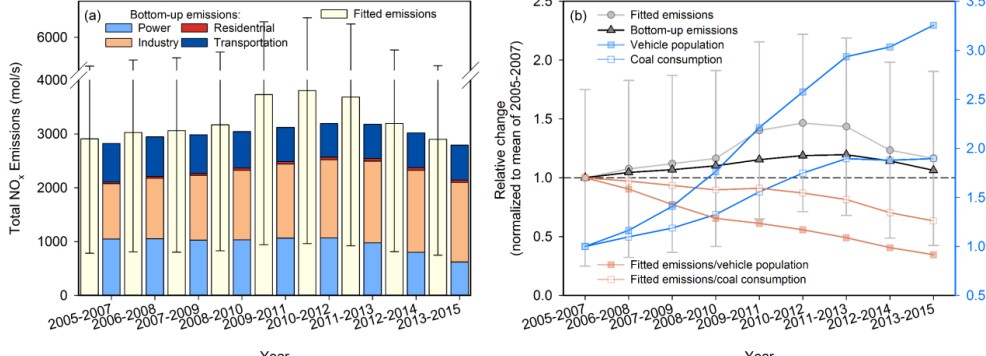

**Figure 3: (a) Fitted (yellow bar) and total anthropogenic NO$_x$ emissions by sector for all investigated cities in this study during 2005 to 2015. The emissions data are derived from the MEIC model. (b) Interannual trends of the fitted (gray line) and bottom-up anthropogenic NO$_x$ emissions for selected cities with valid information on the vehicle population (blue solid squares) and coal consumption (blue open squares) from 2005–2015. The pink lines denote the ratio of the fitted NO$_x$ emissions in this study to the vehicle population (solid squares) and coal consumption (open squares). The relative changes of the vehicle population and coal consumption are indicated by right axes. Error bars show the uncertainties for the fitted emissions by using this method (see Sect. 3.4).**



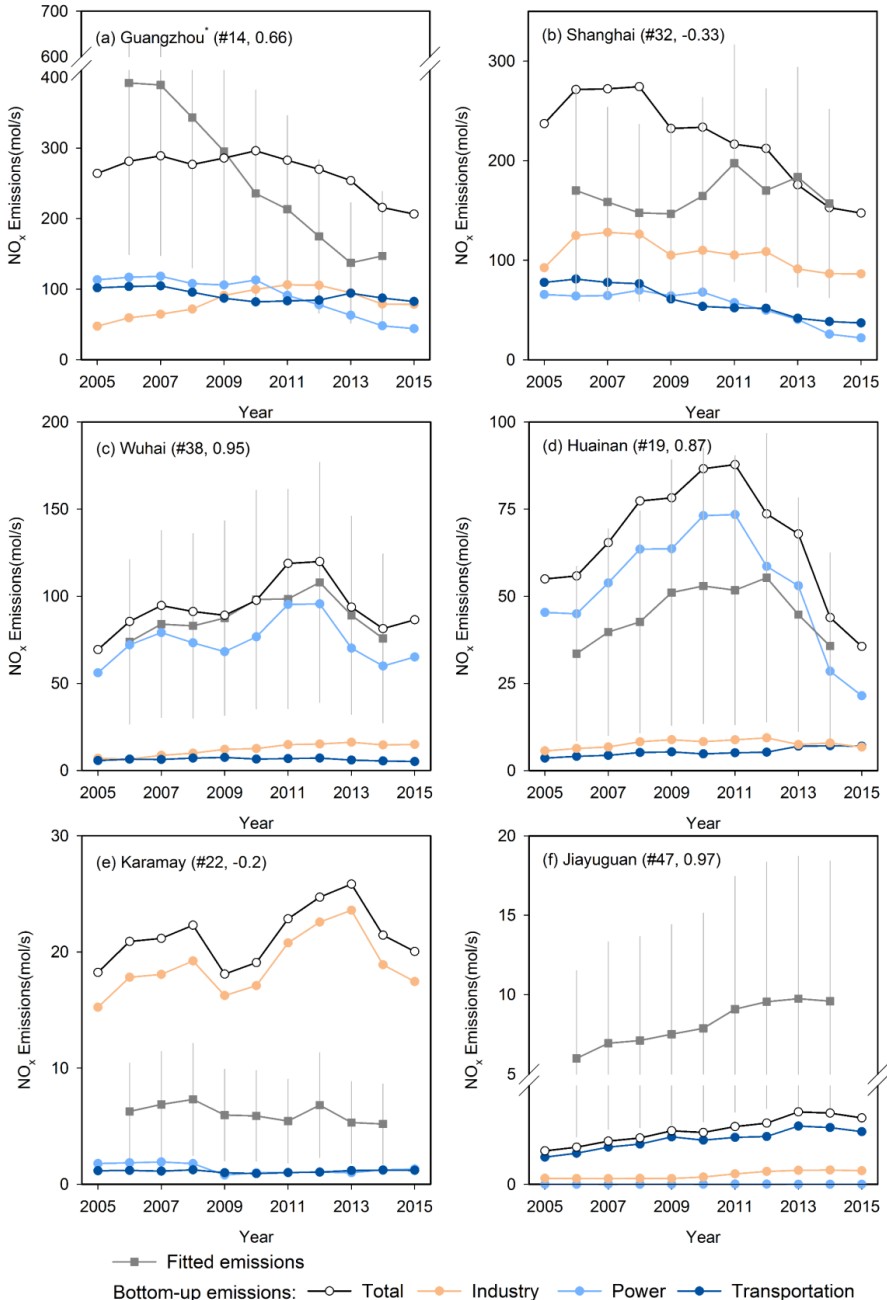

**Figure 4: Interannual trends in the fitted (gray squares) and bottom-up anthropogenic NOₓ emissions for 2005 to 2015 including total (black circles), industrial (pink circles), power plant (light blue circles) and transportation (dark blue circles) emissions. Error bars denote the uncertainty of the fitted NOₓ emissions. The IDs from Fig. 1 and the correlation coefficients of the pair-wise trends between the bottom-up and fitted NOₓ emissions are shown in the bracket after the name of the city.**

Guangzhou* represents the cities of Guangzhou, Foshan and Dongguan, which are recognized as the same hot spot in the map of NO₂ TVCDs.



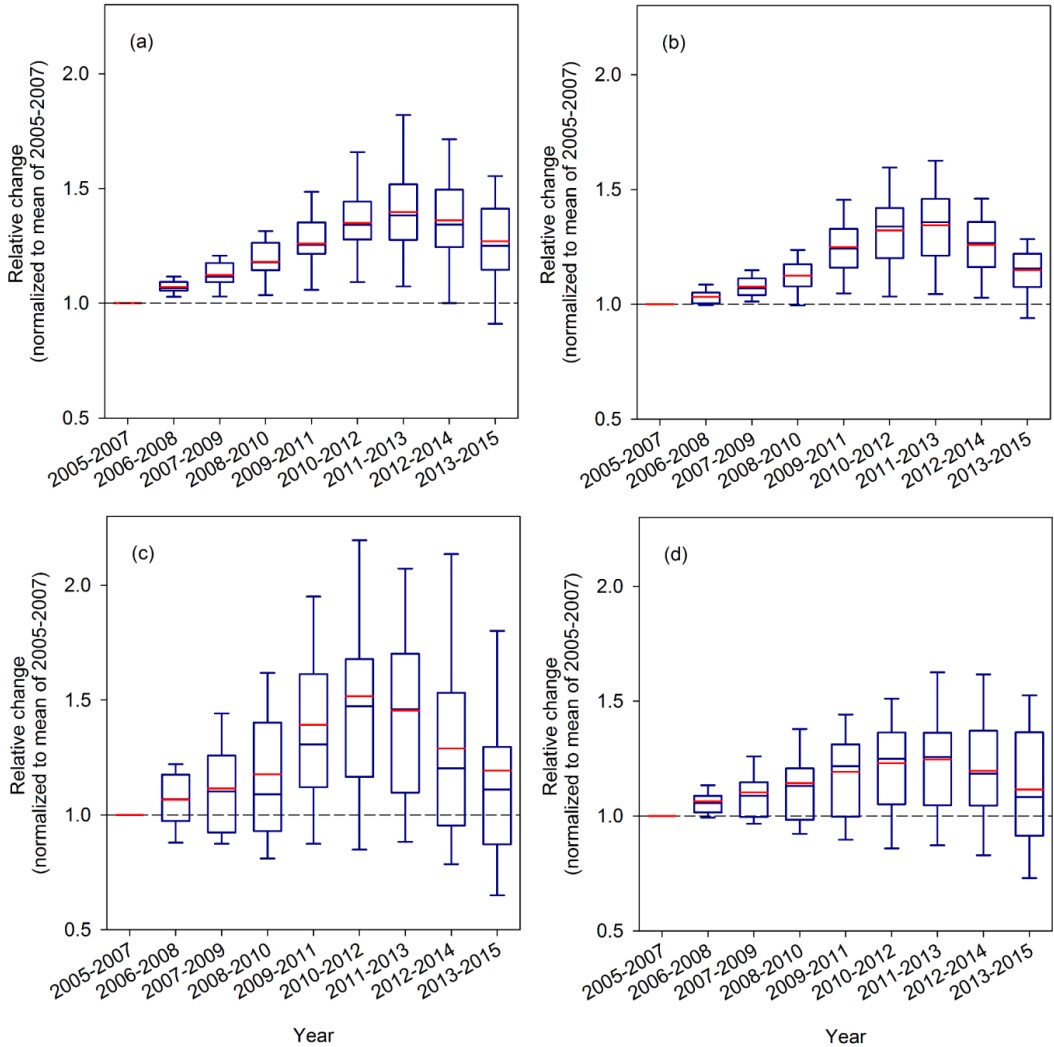

**Figure 5: Comparisons of the trends in satellite observations (left panels) with those in the bottom-up emission inventory (right panels) at the province and city level during 2005 to 2015. The box plots show the relative changes in (a) the average OMI tropospheric NO₂ column densities for provinces in China; (b) the anthropogenic NOₓ emissions for provinces in China; (c) the fitted NOₓ emissions for cities investigated in this study; and (d) the anthropogenic NOₓ emissions for the corresponding cities. The blue horizontal line is the median of the relative differences; the red horizontal line is the mean of the relative differences; the box denotes the 25% and 75% percentiles; and the whiskers denote the 10% and 90% percentiles. The bottom-up emission data are derived from the MEIC model.**





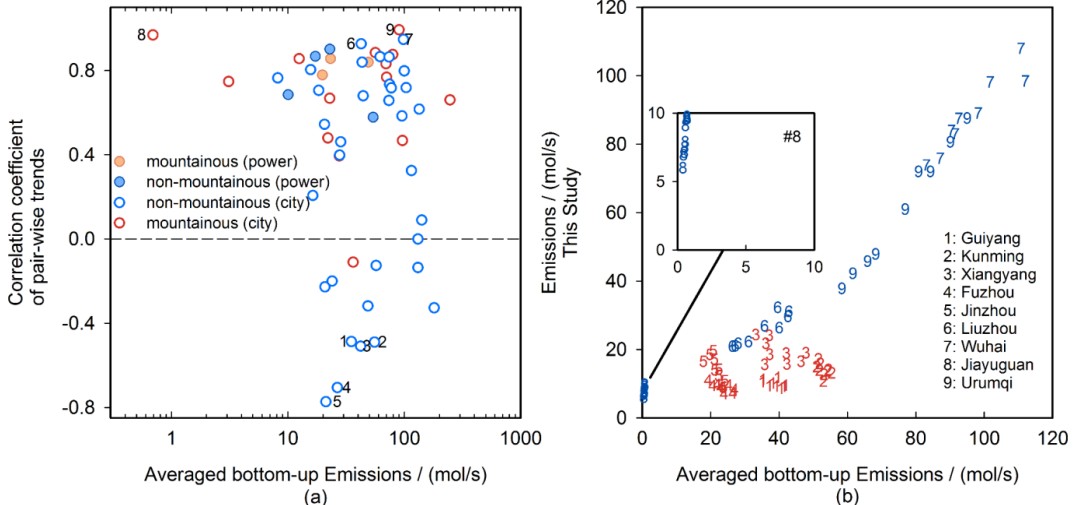

(a)        (b)

**Figure 6: (a) Correlation coefficients of the pair-wise trends between the bottom-up and fitted NOₓ emissions for all selected sites during 2006\* to 2014\*. The results for sites with correlation coefficients less than -0.4 or larger than 0.9 are indicated by digits. (b) Scatterplots of the fitted NOₓ emissions for investigated cities versus bottom-up emission inventories during 2006\* to 2014\*. Urban emissions from bottom-up inventories are integrated over an area of 40 km × 40 km (see Sect. 2.2). Results with correlation coefficients less than -0.4 or larger than 0.9 are color coded by red and blue, respectively.**

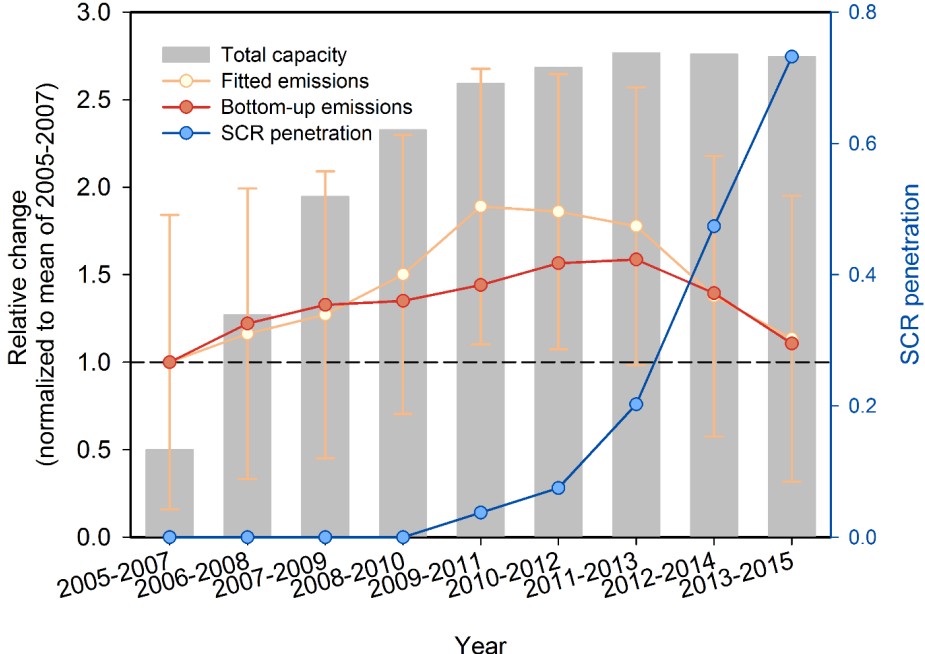

**Figure 7: Interannual trends of the fitted (red line) and bottom-up (pink line) NOₓ emissions for selected power plants during 2005 to 2015. The bar denotes the total capacity of selected power plants. The blue line denotes the penetration of power plants with denitration devices. Error bars show the uncertainties for fitted emissions by this method (see Sect. 3.4).**





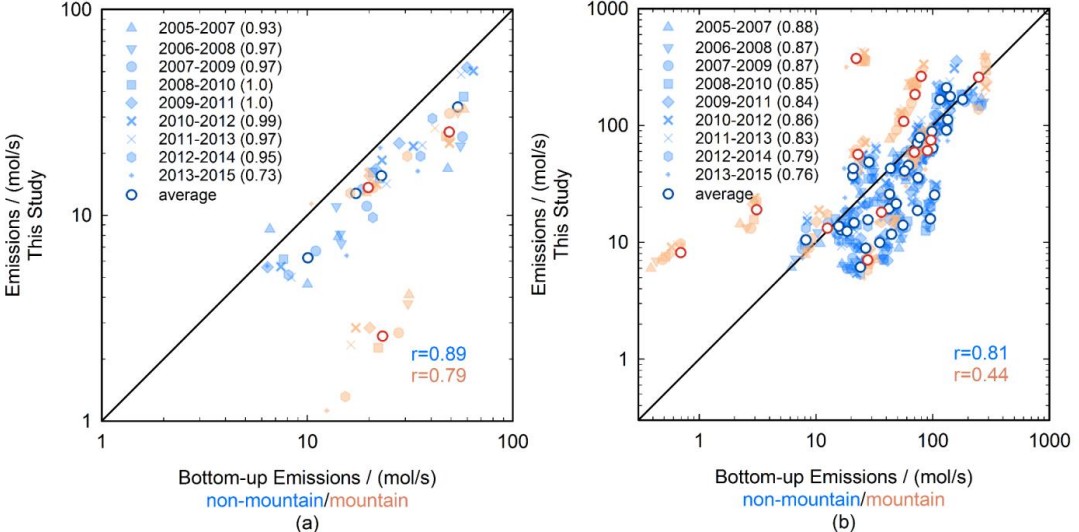

**Figure 8: Scatterplots of the fitted NO$_x$ emissions for the investigated (a) power plants and (b) cities versus the bottom-up emission inventories during 2006\* to 2014\*. Urban emissions from bottom-up inventories are integrated over an area of 40 km × 40 km (see Sect. 2.2). The correlation coefficients of non-mountainous sites for individual 3-year periods**
5 **are shown in brackets. Open circles represent the average emissions for non-mountainous (blue) and mountainous (red) sites during the entire period. The correlation coefficients of the average emissions for non-mountainous and mountainous sites are color-coded in blue and red, respectively. The straight line represents the ratio of 1:1.**