# Peer review of "$NO_x$ emission trends over Chinese cities estimated from OMI observations during 2005 to 2015"

_Atmospheric Chemistry and Physics, 2017_

## Referee Comment (RC1) · Anonymous Referee #1 · 19 May 2017

Review of the manuscript: "NOx emission trends over Chinese cities estimated from OMI observations during 2005 to 2015" by Liu et al.

The manuscript presents a method to determine trends in NOx emission over China. The authors apply a methodology, introduced by the same authors in a previous paper, to determine NOx emission from satellite-based observations. The approach is particularly valuable as it is independent of chemical transport models and their uncertainty/assumptions. The results confirm the observed decline in the Chinese NOx emissions after year 2011. I recommend the publication after addressing the following comments:

Specific comments

1. Several recent studies have shown decreasing NOx levels in China from satellite

data. Can you evaluate how your trends compare with these existing results? This is mentioned in the introduction but it could be discussed in the conclusion too or where you present your numerical results? The results are derived at different resolutions I guess, but are you able to evaluate how consistent they are? For example, in this manuscript (Recent reduction in NOx emissions over China: synthesis of satellite observations and emission inventories doi:10.1088/1748-9326/11/11/114002) you analysed the NO2 peak year: how do the peak year for the provinces agrees with your latest city level results? Answering this question you should also be able to stress the added value of this work, compared to existing results.

2. Section 2.1 and later: You talk about "valid lifetime" or "satisfactory result" for the fitting: could you remind the reader how you define a satisfactory fitting? Especially for the power plants (only 7 good ones) can you explain the reasons for the unsuccessful fits?

3. Fig. 7 and page 8: What do you mean by market share of SCR? Share with respect to what? Could you define that?

4. Fig. 8 Can you comment on why for power plants there is a sort of bias, with bottom-up emissions generally higher than your emissions? (All points are below the 1:1 line)

5. Section 3.4 What kind a error/bias is due to the fact that you use summer days and clear sky data? How do you see this might affect your comparison with bottom-up inventories?

Technical comments

6. Figure 6 Please specify in the caption that you mean anthropogenic as bottom-up inventory, the emission you calculate by fitting are also anthropogenic, they might get confused. Also the color coding in confusing, could you use something else than red-blue in b-panel, because one might thing they relate to the red-blue of panel a, while

they are not.

---

## Referee Comment (RC2) · Anonymous Referee #2 · 29 May 2017

China is experiencing dramatic changes in economy and energy structure. Due to the poor air quality in developed regions, a series of emission control measures have been taken and changes in air pollutant emissions could be expected. Independent from the bottom-up emission inventory that might be limited by the accuracy and timeliness of data, this work applied OMI NO2 data and estimated the trends of NOX emissions for selected cities and power plants across the country, following previous studies from the same research group. In general the paper was well organized, clearly written, and easy to follow. I recommend its publication with some more discussions or corrections as suggested below:

1. Is there any big difference or correction in the method of "top-down" emission calculation between this work and the authors' previous studies (i.e., Liu et al., 2016a; b)? I understand the inter-annual trend is included in this work, but other improvement in

method (if any) needs to be clarified so that the audience could easily compare different papers.

2. Pages 4-5, the authors said they presented cities/plants with satisfactory fitting results. Here needs some explanations: what's the criterion of examining the fitting results, and why were there any "unsatisfactory results"? Does that imply that there are some problems or limitations in the calculation method and it cannot be applied for all the selected cities/plants?

3. Although the emission trends between top-down and bottom-up methods were generally consistent with each other, it seems that larger emission growth was estimated based on the OMI data than MEIC for both cities and power plants before 2012 (e.g., Figure 3 and 7). Uncertainties in bottom-up emissions (i.e., MEIC) might be part of the reasons, and I suggest a paragraph of discussion including comparisons with other available bottom-up estimates.

4. Figure 4. Did that imply the poorer estimation in emissions from small industrial plants than those from power plants in MEIC? Needs clarification.

5. Small issues: Line 40, Page 4: "section 2.2" ? please check. Line 28, Page 5: "new dry-process" or "precalciner"?

---

## Author Comment (AC1) · 23 Jun 2017

*The manuscript presents a method to determine trends in $NO_x$ emission over China. The authors apply a methodology, introduced by the same authors in a previous paper, to determine $NO_x$ emission from satellite-based observations. The approach is particularly valuable as it is independent of chemical transport models and their uncertainty/assumptions. The results confirm the observed decline in the Chinese $NO_x$ emissions after year 2011. I recommend the publication after addressing the following comments:*

**Response:** We thank Referee #1 for the encouraging comments. All comments and suggestions have been considered carefully and well addressed below.

*Specific comments:*
*1. Several recent studies have shown decreasing $NO_x$ levels in China from satellite data. Can you evaluate how your trends compare with these existing results? This is mentioned in the introduction but it could be discussed in the conclusion too or where you present your numerical results? The results are derived at different resolutions I guess, but are you able to evaluate how consistent they are? For example, in this manuscript (Recent reduction in $NO_x$ emissions over China: synthesis of satellite observations and emission inventories doi:10.1088/1748-9326/11/11/114002) you analyzed the $NO_2$ peak year: how do the peak year for the provinces agrees you're your latest city level results? Answering this question you should also be able to stress the added value of this work, compared to existing results.*

**Response:** We thank for the suggestion and add the discussion of the comparison with other existing studies to the conclusion, as follows:

"The average emission trend fitted by this study is consistent with the previous findings, which showed that OMI $NO_2$ levels peaked in 2011 over China (Krotkov et al., 2016; Duncan et al., 2016) and $NO_x$ emissions from satellite data assimilation peaked in 2011/2012 (Miyazaki et al., 2017; van der A et al., 2017; Souri et al., 2017) respectively. Additionally, the fitted emission peaks for individual cities showed reasonable agreement with the peaks of OMI $NO_2$ levels at provincial level (Liu et al., 2016b). Half of the investigated cities reached simultaneous emission peaks with the corresponding provinces. For the another half, the majority (over 70%) reached emission peaks prior to the average provincial timeline, which are most likely caused by emission control policies implemented in the city ahead of the provincial schedule, such as the previously discussed new vehicle emission standards in Guangzhou."

*2. Section 2.1 and later: You talk about "valid lifetime" or "satisfactory result" for the fitting: could you remind the reader how you define a satisfactory fitting? Especially for the power plants (only 7 good ones) can you explain the reasons for the unsuccessful fits?*

**Response:** We followed the criteria defined in Liu et al. (2016a) to assure a good fit performance (i.e., satisfactory fitting). We add the description for the criteria in Section 2.1, as follows:

"The fitting results with poor performance (i.e., R<0.9, lower bound of confidence interval CI <0, CI width for lifetime >10 h, CI width for the $NO_2$ mass >0.8×mass) were discarded, in accordance with the criteria in Sect. 2.2 of Liu et al. (2016a)."

We failed to get the satisfactory fitting results for most power plants because their signals are not strong enough to be distinguished from the surroundings, particularly for those located in/near urban areas. More than half of the power plants were discarded from the final analysis because they locate in a radius of 100 km around city centers. Others were dismissed due to the low R or unreasonable CI resulting from the low signal/background ratio. The number of power plants with valid fitting results decreases sharply when $NO_2$ concentrations over power plants decline because of the installation of denitration devices. The number of power plants with satisfactory results for the period of 2013-2015 is only half of that for the period of 2005-2007.

We have rephrased the sentences in Section 2.3, as follows:

"Among over 200 pre-selected cities, 48 cities (including 14 mountainous sites) were fitted with good performance (see the definition in Sect.2.1). While among over 100 pre-selected power plants, more than half were excluded from the fit procedure, because they are located in a radius of 100 km around prefecture-level city centers, on the basis of a visual inspection of satellite imagery from Google Earth. Only 7 power plants (including 3 mountainous sites) were fitted with good performance."

*3. Fig. 7 and page 8: What do you mean by market share of SCR? Share with respect to what? Could you define that?*

**Response:** We define the market share of SCR as the percentage of unit capacity of power plants installing SCR in the total capacity of all the power plants. We replaced "market share" with "penetration", which is more commonly used in the emission inventory community, and added the definition in the revised manuscript.

*4. Fig. 8 Can you comment on why for power plants there is a sort of bias, with bottom-up emissions generally higher than your emissions? (All points are below the 1:1 line)*

**Response:** We agree that there are certain uncertainties of the fitted emissions, which are explained in detail in Section 3.4. Emissions for mountainous sites are expected to be biased due to the bias in wind fields (Liu et al., 2016a). But we do not expect a systematic bias associated with those uncertainties for non-mountainous sites. This is confirmed by the comparison for power plants with valid fitting results for the period of 2010-2012 in the figure S1. There are differences between the fitted and bottom-up estimates, but no significant bias. It is thus probably coincidence that the fitted emissions for the four power plants (blue points in Fig. 8a & S1) that have valid fitting results for each three consecutive years from 2005 to 2015 are lower than the bottom-up estimates.

[Figure]

Figure S1: Scatterplots of the fitted $NO_x$ emissions for the investigated non-mountainous power plants versus the bottom-up emission inventories (MEIC) during 2010 to 2012. The sites displayed in Fig.8a are color coded by blue.

*5. Section 3.4 What kind a error/bias is due to the fact that you use summer days and clear sky data? How do you see this might affect your comparison with bottom-up inventories?*

**Response:** Concerning the usage of summer data, we generally agree that there are monthly variations in $NO_x$ emissions for cities and power plants in China. Emissions typically peak in December of each year because of high year-end industrial activities (Li et al., 2017). Thus, the fitted emission rates based on non-winter satellite data maybe biased compared to the annual mean rates. However, it will not affect the comparison with bottom-up inventories, because only bottom-up emissions for non-winter seasons were used for comparison (see Section 2.2).

With respect to clear sky data, we agree that the selection of cloud-free OMI $NO_2$ TVCDs used for fitting emissions does not represent the average level for all days, due to the accelerated photochemistry and different meteorological conditions (e.g. boundary layer height, atmospheric transport) under clear sky conditions. But still the emission estimates are appropriate, as both the $NO_x$ lifetime and total mass derived from the $NO_2$ TVCDs are derived consistently, both of which reflect the values under clear sky conditions. Thus, this effect is of minor importance for this study and is not expected to bias the estimated $NO_x$ emissions.

*Technical comments*
*6. Figure 6 Please specify in the caption that you mean anthropogenic as bottom-up inventory, the emission you calculate by fitting are also anthropogenic, they might get confused. Also the color coding in confusing, could you use something else than redblue in b-panel, because one might thing they relate to the red-blue of panel a, while they are not.*

**Response:** Thanks. We have specified it in the caption and changed the color to green/grey in Figure 6 in the revised manuscript.

**References**

Duncan, B. N., Lamsal, L. N., Thompson, A. M., Yoshida, Y., Lu, Z., Streets, D. G., Hurwitz, M. M., and Pickering, K. E.: A space-based, high-resolution view of notable changes in urban $NO_x$ pollution around the world (2005–2014), J. Geophys. Res., 121, 976–996, doi: 10.1002/2015jd024121, 2016.

Krotkov, N. A., McLinden, C. A., Li, C., Lamsal, L. N., Celarier, E. A., Marchenko, S. V., Swartz, W. H., Bucsela, E. J., Joiner, J., Duncan, B. N., Boersma, K. F., Veefkind, J. P., Levelt, P. F., Fioletov, V. E., Dickerson, R. R., He, H., Lu, Z., and Streets, D. G.: Aura OMI observations of regional $SO_2$ and $NO_2$ pollution changes from 2005 to 2015, Atmos. Chem. Phys., 16, 4605–4629, doi: 10.5194/acp-16-4605-2016, 2016.

Li, M., Zhang, Q., Kurokawa, J. I., Woo, J. H., He, K., Lu, Z., Ohara, T., Song, Y., Streets, D. G., Carmichael, G. R., Cheng, Y., Hong, C., Huo, H., Jiang, X., Kang, S., Liu, F., Su, H., and Zheng, B.: MIX: a mosaic Asian anthropogenic emission inventory under the international collaboration framework of the MICS-Asia and HTAP, Atmos. Chem. Phys., 17, 935–963, doi: 10.5194/acp-17-935-2017, 2017.

Liu, F., Beirle, S., Zhang, Q., Dörner, S., He, K., and Wagner, T.: $NO_x$ lifetimes and emissions of cities and power plants in polluted background estimated by satellite observations, Atmos. Chem. Phys., 16, 5283–5298, doi: 10.5194/acp-16-5283-2016, 2016a.

Liu, F., Zhang, Q., Ronald, J. van der A., Zheng, B., Tong, D., Yan, L., Zheng, Y., and He, K.: Recent reduction in $NO_x$ emissions over China: synthesis of satellite observations and emission inventories, Environmental Research Letters, 11, 114002, 2016b.

Miyazaki, K., Eskes, H., Sudo, K., Boersma, K. F., Bowman, K., and Kanaya, Y.: Decadal changes in global surface $NO_x$ emissions from multi-constituent satellite data assimilation, Atmos. Chem. Phys., 17, 807–837, doi: 10.5194/acp-17-807-2017, 2017.

Souri, A. H., Choi, Y., Jeon, W., Woo, J.-H., Zhang, Q., and Kurokawa, J.-i.: Remote sensing evidence of decadal changes in major tropospheric ozone precursors over East Asia, J. Geophys. Res., 122, 2474–2492, doi: 10.1002/2016JD025663, 2017.

van der A, R. J., Mijling, B., Ding, J., Koukouli, M. E., Liu, F., Li, Q., Mao, H., and Theys, N.: Cleaning up the air: effectiveness of air quality policy for $SO_2$ and $NO_x$ emissions in China, Atmos. Chem. Phys., 17, 1775–1789, doi: 10.5194/acp-17-1775-2017, 2017.

---

## Author Comment (AC2) · 23 Jun 2017

*China is experiencing dramatic changes in economy and energy structure. Due to the poor air quality in developed regions, a series of emission control measures have been taken and changes in air pollutant emissions could be expected. Independent from the bottom-up emission inventory that might be limited by the accuracy and timeliness of data, this work applied OMI $NO_2$ data and estimated the trends of $NO_x$ emissions for selected cities and power plants across the country, following previous studies from the same research group. In general the paper was well organized, clearly written, and easy to follow. I recommend its publication with some more discussions or corrections as suggested below:*

**Response:** We thank Referee #2 for the encouraging comments. We addressed the comments carefully as below.

*Specific comments:*

*1. Is there any big difference or correction in the method of "top-down" emission calculation between this work and the authors' previous studies (i.e., Liu et al., 2016a; b)? I understand the inter-annual trend is included in this work, but other improvement in method (if any) needs to be clarified so that the audience could easily compare different papers.*

**Response:** In general, there is no fundamental change in methodology between this study and our previous one (Liu et al., 2016a). In order to perform the inter-annual analysis, we made the adjustment as follows: we based the estimation on $NO_2$ columns around the source of interest averaged over three years, similar as in previous studies (e.g., Fioletov et al., 2011; Lu et al., 2015); afterwards, the total $NO_x$ mass was integrated over the mean TVCDs at weak wind speeds (below 3 m/s) instead of calm winds (below 2 m/s) used in our previous work for better statistics. We have clarified this in Sect.2.1.

Another previous study (Liu et al., 2016b) calculated changes in OMI $NO_2$ column densities for each province over China from 2005 to 2015 and compared them with the bottom-up inventory to examine $NO_x$ emission trends and their driving forces. It is not directly associated with the methodology in this study.

*2. Pages 4-5, the authors said they presented cities/plants with satisfactory fitting results. Here needs some explanations: what's the criterion of examining the fitting results, and why were there any "unsatisfactory results"? Does that imply that there are some problems or limitations in the calculation method and it cannot be applied for all the selected cities/plants?*

**Response:** We followed the criteria defined in Liu et al. (2016a) to assure a good fit performance. We add the description for the criteria in Section 2.1, as follows:

"The fitting results with poor performance (i.e., R<0.9, lower bound of confidence interval CI <0, CI width for lifetime >10 h, CI width for the $NO_2$ mass >0.8×mass) were discarded, in accordance with the criteria in Sect. 2.2 of Liu et al. (2016a)."

Satisfactory results stand for fitting results meeting the criteria above. The fit method

can be applied for all cities/plants in principle, but the fit may fail because of the low R or unreasonable CI resulting from the low signal/background ratio. We rephrase the sentences in Section 2.3, as follows:

"Among over 200 pre-selected cities, 48 cities (including 14 mountainous sites) were fitted with good performance (see the definition in Sect.2.1). While among over 100 pre-selected power plants, over half were excluded from the fit procedure, because they are located in a radius of 100 km around prefecture-level city centers, on the basis of a visual inspection of satellite imagery from Google Earth. Only 7 power plants (including 3 mountainous sites) were fitted with good performance."

*3. Although the emission trends between top-down and bottom-up methods were generally consistent with each other, it seems that larger emission growth was estimated based on the OMI data than MEIC for both cities and power plants before 2012 (e.g., Figure 3 and 7). Uncertainties in bottom-up emissions (i.e., MEIC) might be part of the reasons, and I suggest a paragraph of discussion including comparisons with other available bottom-up estimates.*

**Response:** We agree that uncertainties in bottom-up emissions contribute to the differences, which has been detailed in Section 3.2. The spatial allocation approach adopted in bottom-up inventories tends to diminish regional diversity and may consequently result in smaller emission growth compared to top-down estimates. This is a general limitation of applying regional inventories to calculating urban emissions. We thank for the suggestion of including more bottom-up inventories into the comparison. We added the comparison with other widely used bottom-up emission inventories in the community. In order to keep consistent with the top-down estimates derived from 3-year average $NO_2$ concentrations, only inventories with more than three years data available from 2005 to 2015 are included. We finally added Emission Database for Global Atmospheric Research version 4.3 (EDGAR v4.3, Crippa et al. 2016), Regional Emission inventory in Asia version 2.1 (REAS v2.1, Kurokawa et al., 2013). All bottom-up inventories show similar growth rates, suggesting that the differences to our estimates indeed arise from the methodology of deriving urban emissions from regional inventories, rather than depend on the chosen inventory. We added a brief introduction of the available bottom-up inventories in Section 2.2 and further discussed the comparison with the fitting results in Section 3.1.

*4. Figure 4. Did that imply the poorer estimation in emissions from small industrial plants than those from power plants in MEIC? Needs clarification.*

**Response:** We agree that it implies the poorer estimation in emissions from small industrial plants that those from power plants in the MEIC inventory. We have clarified it in Section 3.2, as follows:

"For industrial emissions, MEIC first downscaled provincial totals to counties using industrial GDP, and then allocate county emissions to grids with population density. Thus uncertainty of emissions from the industrial sector is larger than that from power plants."

*5. Small issues: Line 40, Page 4: "section 2.2" ? please check. Line 28, Page 5: "new dry-process" or "precalciner"?*

**Response:** Thanks for pointing out them. We have replaced "section 2.2" with "section 2.1" and replaced "new dry-process" with "precalciner" in the revised manuscript.

**References**

Crippa, M., Janssens-Maenhout, G., Dentener, F., Guizzardi, D., Sindelarova, K., Muntean, M., Van Dingenen, R., and Granier, C.: Forty years of improvements in European air quality: regional policy-industry interactions with global impacts, Atmos. Chem. Phys., 16, 3825–3841, 10.5194/acp-16-3825-2016, 2016.

Fioletov, V. E., McLinden, C. A., Krotkov, N., Moran, M. D., and Yang, K.: Estimation of $SO_2$ emissions using OMI retrievals, Geophys. Res. Lett., 38, L21811, doi: 10.1029/2011gl049402, 2011.

Kurokawa, J., Ohara, T., Morikawa, T., Hanayama, S., Janssens-Maenhout, G., Fukui, T., Kawashima, K., and Akimoto, H.: Emissions of air pollutants and greenhouse gases over Asian regions during 2000–2008: Regional Emission inventory in ASia (REAS) version 2, Atmos. Chem. Phys., 13, 11019–11058, doi: 10.5194/acp-13-11019-2013, 2013.

Liu, F., Beirle, S., Zhang, Q., Dörner, S., He, K., and Wagner, T.: $NO_x$ lifetimes and emissions of cities and power plants in polluted background estimated by satellite observations, Atmos. Chem. Phys., 16, 5283–5298, doi: 10.5194/acp-16-5283-2016, 2016a.

Liu, F., Zhang, Q., Ronald, J. van der A., Zheng, B., Tong, D., Yan, L., Zheng, Y., and He, K.: Recent reduction in $NO_x$ emissions over China: synthesis of satellite observations and emission inventories, Environmental Research Letters, 11, 114002, 2016b.

Lu, Z., Streets, D. G., de Foy, B., and Krotkov, N. A.: Ozone Monitoring Instrument observations of interannual increases in $SO_2$ emissions from Indian coal-fired power plants during 2005–2012, Environ. Sci. Technol., 47, 13993–14000, 2013.

Lu, Z., Streets, D. G., de Foy, B., Lamsal, L. N., Duncan, B. N., and Xing, J.: Emissions of nitrogen oxides from US urban areas: estimation from Ozone Monitoring Instrument retrievals for 2005–2014, Atmos. Chem. Phys., 15, 10367–10383, doi: 10.5194/acp-15-10367-2015, 2015.

---

## Author Response (AR1)

*The manuscript presents a method to determine trends in $NO_x$ emission over China. The authors apply a methodology, introduced by the same authors in a previous paper, to determine $NO_x$ emission from satellite-based observations. The approach is particularly valuable as it is independent of chemical transport models and their uncertainty/assumptions. The results confirm the observed decline in the Chinese $NO_x$ emissions after year 2011. I recommend the publication after addressing the following comments:*

**Response:** We thank Referee #1 for the encouraging comments. All comments and suggestions have been considered carefully and well addressed below.

*Specific comments:*
*1. Several recent studies have shown decreasing $NO_x$ levels in China from satellite data. Can you evaluate how your trends compare with these existing results? This is mentioned in the introduction but it could be discussed in the conclusion too or where you present your numerical results? The results are derived at different resolutions I guess, but are you able to evaluate how consistent they are? For example, in this manuscript (Recent reduction in $NO_x$ emissions over China: synthesis of satellite observations and emission inventories doi:10.1088/1748-9326/11/11/114002) you analyzed the $NO_2$ peak year: how do the peak year for the provinces agrees you're your latest city level results? Answering this question you should also be able to stress the added value of this work, compared to existing results.*

**Response:** We thank for the suggestion and add the discussion of the comparison with other existing studies to the conclusion, as follows:

"The average emission trend fitted by this study is consistent with the previous findings, which showed that OMI $NO_2$ levels peaked in 2011 over China (Krotkov et al., 2016; Duncan et al., 2016) and $NO_x$ emissions from satellite data assimilation peaked in 2011/2012 (Miyazaki et al., 2017; van der A et al., 2017; Souri et al., 2017) respectively. Additionally, the fitted emission peaks for individual cities showed reasonable agreement with the peaks of OMI $NO_2$ levels at provincial level (Liu et al., 2016b). Half of the investigated cities reached simultaneous emission peaks with the corresponding provinces. For the another half, the majority (over 70%) reached emission peaks prior to the average provincial timeline, which are most likely caused by emission control policies implemented in the city ahead of the provincial schedule, such as the previously discussed new vehicle emission standards in Guangzhou."

*2. Section 2.1 and later: You talk about "valid lifetime" or "satisfactory result" for the fitting: could you remind the reader how you define a satisfactory fitting? Especially for the power plants (only 7 good ones) can you explain the reasons for the unsuccessful fits?*

**Response:** We followed the criteria defined in Liu et al. (2016a) to assure a good fit performance (i.e., satisfactory fitting). We add the description for the criteria in Section 2.1, as follows:

"The fitting results with poor performance (i.e., R<0.9, lower bound of confidence interval CI <0, CI width for lifetime >10 h, CI width for the $NO_2$ mass >0.8×mass) were discarded, in accordance with the criteria in Sect. 2.2 of Liu et al. (2016a)."

We failed to get the satisfactory fitting results for most power plants because their signals are not strong enough to be distinguished from the surroundings, particularly for those located in/near urban areas. More than half of the power plants were discarded from the final analysis because they locate in a radius of 100 km around city centers. Others were dismissed due to the low R or unreasonable CI resulting from the low signal/background ratio. The number of power plants with valid fitting results decreases sharply when $NO_2$ concentrations over power plants decline because of the installation of denitration devices. The number of power plants with satisfactory results for the period of 2013-2015 is only half of that for the period of 2005-2007.

We have rephrased the sentences in Section 2.3, as follows:

"Among over 200 pre-selected cities, 48 cities (including 14 mountainous sites) were fitted with good performance (see the definition in Sect.2.1). While among over 100 pre-selected power plants, more than half were excluded from the fit procedure, because they are located in a radius of 100 km around prefecture-level city centers, on the basis of a visual inspection of satellite imagery from Google Earth. Only 7 power plants (including 3 mountainous sites) were fitted with good performance."

*3. Fig. 7 and page 8: What do you mean by market share of SCR? Share with respect to what? Could you define that?*

**Response:** We define the market share of SCR as the percentage of unit capacity of power plants installing SCR in the total capacity of all the power plants. We replaced "market share" with "penetration", which is more commonly used in the emission inventory community, and added the definition in the revised manuscript.

*4. Fig. 8 Can you comment on why for power plants there is a sort of bias, with bottom-up emissions generally higher than your emissions? (All points are below the 1:1 line)*

**Response:** We agree that there are certain uncertainties of the fitted emissions, which are explained in detail in Section 3.4. Emissions for mountainous sites are expected to be biased due to the bias in wind fields (Liu et al., 2016a). But we do not expect a systematic bias associated with those uncertainties for non-mountainous sites. This is confirmed by the comparison for power plants with valid fitting results for the period of 2010-2012 in the figure S1. There are differences between the fitted and bottom-up estimates, but no significant bias. It is thus probably coincidence that the fitted emissions for the four power plants (blue points in Fig. 8a & S1) that have valid fitting results for each three consecutive years from 2005 to 2015 are lower than the bottom-up estimates.

[Figure]

Figure S1: Scatterplots of the fitted $NO_x$ emissions for the investigated non-mountainous power plants versus the bottom-up emission inventories (MEIC) during 2010 to 2012. The sites displayed in Fig.8a are color coded by blue.

*5. Section 3.4 What kind a error/bias is due to the fact that you use summer days and clear sky data? How do you see this might affect your comparison with bottom-up inventories?*

**Response:** Concerning the usage of summer data, we generally agree that there are monthly variations in $NO_x$ emissions for cities and power plants in China. Emissions typically peak in December of each year because of high year-end industrial activities (Li et al., 2017). Thus, the fitted emission rates based on non-winter satellite data maybe biased compared to the annual mean rates. However, it will not affect the comparison with bottom-up inventories, because only bottom-up emissions for non-winter seasons were used for comparison (see Section 2.2).

With respect to clear sky data, we agree that the selection of cloud-free OMI $NO_2$ TVCDs used for fitting emissions does not represent the average level for all days, due to the accelerated photochemistry and different meteorological conditions (e.g. boundary layer height, atmospheric transport) under clear sky conditions. But still the emission estimates are appropriate, as both the $NO_x$ lifetime and total mass derived from the $NO_2$ TVCDs are derived consistently, both of which reflect the values under clear sky conditions. Thus, this effect is of minor importance for this study and is not expected to bias the estimated $NO_x$ emissions.

*Technical comments*

*6. Figure 6 Please specify in the caption that you mean anthropogenic as bottom-up inventory, the emission you calculate by fitting are also anthropogenic, they might get confused. Also the color coding in confusing, could you use something else than redblue in b-panel, because one might thing they relate to the red-blue of panel a, while they are not.*

**Response:** Thanks. We have specified it in the caption and changed the color to green/grey in Figure 6 in the revised manuscript.

We selected the Huolin power plant (site 2#, 45.5°N, 119.7°E), which is located in Holingol, a county-level city of Inner Mongolia of China (shown in Fig. 1), to demonstrate our approach. The Huolin power plant has a total capacity of 2400 MW

10  and dominates the $NO_x$ emissions from the city of Holingol, contributing over 80% of the total emissions estimated by using the Multi-resolution Emission Inventory for China (see Sect. 2.2), which is a bottom-up emission inventory. Fig. 2(a) displays the 3-year average $NO_2$ TVCDs around the power plant under weak-wind conditions from 2005 to 2015. For simplicity, the 3-year period is represented by the middle year with an asterisk (e.g., 2006* denotes the period from 2005 to 2007). A significant increase in TVCDs was observed from 2006* to 2010*, which was followed by a subsequent decrease.

15  Fig. 2(b) presents the fitted background and $NO_x$ emissions. The fitted $NO_x$ emissions showed an increase of up to a factor of four from 2006* to 2010* and a decrease of 30% from 2011* to 2014*, whereas the fitted background was steady and showed a standard deviation of less than 10% from 2006* to 2014*. The growth of the fitted $NO_x$ emissions in the early stage was found to be consistent with the construction of new electric-generating units, with the total capacity increasing from 300 MW to 2400 MW from 2005 to 2009. Subsequently, the fitted $NO_x$ emissions remained steady from 2010* to 2012*, when no

20  new electric-generating units were placed into service, and finally decreased after the installation of Selective Catalytic Reduction (SCR) equipment at the power plants. This decrease in emissions indicated the effectiveness of SCR equipment for decreasing emissions.

**2.2 Bottom-up information**

We used bottom-up information to pre-select promising sites and to perform a comparison with the fitted top-down emission

25  trends. We selected bottom-up emission inventories widely used in the community, in which multi-year gridded estimates are provided (more than three years data available from 2005 to 2015). We finally included Emission Database for Global Atmospheric Research version 4.3 (EDGAR v4.3, available for 1970–2010, Crippa et al. 2016), Regional Emission inventory in Asia version 2.1 (REAS v2.1, available for 2000–2008, Kurokawa et al., 2013) and MEIC (http://www.meicmodel.org) compiled by Tsinghua University. The analysis was focused on the MEIC inventory that are

30  available for the whole period.  Vehicle population and coal consumption at the city level were derived from the China Statistical Yearbook for Regional Economy (NBS: CSYRE, 2004–2014) and the China Environment Yearbook (NBS: CEY, 2004–2015), respectively. We derived the information for the coordinates, unit capacities and technologies  for individual power plants from the unit-based China coal-fired Power plant Emissions Database (CPED) (Liu et al., 2015) integrated in

35  MEIC.

We calculated the $NO_x$ emissions from cities and power plants  from 2005 to 2015. Only emissions for non-winter seasons were considered, in accordance with the emissions included for the top-down estimates, except for EDGAR in which only annual emissions are available. The gridded  bottom-up inventory was integrated over a $40 \times 40$ km$^2$ metropolitan area for which the proposed top-down method was sensitive

40  to calculate the total urban emissions (Liu et al., 2016a). Emissions for individual power plants are derived from CPED and the power plant sector of REAS directly (emissions from individual point sources are not available in EDGAR). Notably, the emissions uncertainties associated with power plants derived from CPED were much lower (30%) than those for cities

[revised manuscript text omitted]

25 previous findings, which showed that OMI $NO_2$ levels peaked in 2011 over China (Krotkov et al., 2016; Duncan et al., 2016) and $NO_x$ emissions from satellite data assimilation peaked in 2011/2012 (Miyazaki et al., 2017; van der A et al., 2017; Souri et al., 2017) respectively. Additionally, the fitted emission peaks for individual cities showed reasonable agreement with the peaks of OMI $NO_2$ levels at provincial level (Liu et al., 2016b). Half of the investigated cities reached simultaneous emission peaks with the corresponding provinces. For the another half, the majority (over 70%) reached emission peaks prior to the

30 average provincial timeline, which are most likely caused by emission control policies implemented in the city ahead of the provincial schedule, such as the previously discussed new vehicle emission standards in Guangzhou.

[revised manuscript text omitted]

the total capacity of all the power plants). **Error bars show the uncertainties for fitted emissions by this method (see Sect. 3.4).**

[Figure]

**Figure 8: Scatterplots of the fitted NO$_x$ emissions for the investigated (a) power plants and (b) cities versus the bottom-up emission inventories during 2006\* to 2014\*. Urban emissions from bottom-up inventories are integrated over an area of 40 km × 40 km (see Sect. 2.2). The correlation coefficients of non-mountainous sites for individual 3-year periods are shown in brackets. Open circles represent the average emissions for non-mountainous (blue) and mountainous (red) sites during the entire period. The correlation coefficients of the average emissions for non-mountainous and mountainous sites are color-coded in blue and red, respectively. The straight line represents the ratio of 1:1.**